# Incidence and Prognostic Factors of Painful Vertebral Compression Fracture Caused by Spine Stereotactic Body Radiotherapy

**DOI:** 10.3390/jcm12113853

**Published:** 2023-06-05

**Authors:** Kei Ito, Kentaro Taguchi, Yujiro Nakajima, Hiroaki Ogawa, Shurei Sugita, Keiko Nemoto Murofushi

**Affiliations:** 1Division of Radiation Oncology, Department of Radiology, Tokyo Metropolitan Cancer and Infectious Diseases Center, Komagome Hospital, 3-18-22 Honkomagome, Bunkyo-ku, Tokyo 113-8677, Japan; k.taguchi.0122@gmail.com (K.T.); ynakajim@komazawa-u.ac.jp (Y.N.); hiroaki.ogawa.b6@tohoku.ac.jp (H.O.); kmurofushi0918@gmail.com (K.N.M.); 2Department of Radiological Sciences, Komazawa University, 1-23-1 Komazawa, Setagaya-ku, Tokyo 154-8525, Japan; 3Department of Radiation Oncology, Tohoku University Hospital, 1-1 Seiryo-machi, Aoba-ku, Sendai 980-8574, Japan; 4Department of Orthopedics, Tokyo Metropolitan Cancer and Infectious Diseases Center, Komagome Hospital, 3-18-22 Honkomagome, Bunkyo-ku, Tokyo 113-8677, Japan; ssugita-tky@umin.ac.jp

**Keywords:** spine SBRT, spinal metastasis, painful VCF, adverse effects, risk factors

## Abstract

Most studies of vertebral compression fractures (VCF) caused by stereotactic body radiotherapy (SBRT) do not discuss the symptoms of this complication. In this paper, we aimed to determine the rate and prognostic factors of painful VCF caused by SBRT for spinal metastases. Spinal segments with VCF in patients treated with spine SBRT between 2013 and 2021 were retrospectively reviewed. The primary endpoint was the rate of painful VCF (grades 2–3). Patient demographic and clinical characteristics were evaluated as prognosticators. In total, 779 spinal segments in 391 patients were analyzed. The median follow-up after SBRT was 18 (range: 1–107) months. Sixty iatrogenic VCFs (7.7%) were identified. The rate of painful VCF was 2.4% (19/779). Eight (1.0%) VCFs required surgery for internal fixation or spinal canal decompression. The painful VCF rate was significantly higher in patients with no posterolateral tumor involvement than in those with bilateral or unilateral involvement (50% vs. 23%; *p* = 0.042); it was also higher in patients with spine without fixation than in those with fixation (44% vs. 0%; *p* < 0.001). Painful VCFs were confirmed in only 2.4% of all the irradiated spinal segments. The absence of posterolateral tumor involvement and no fixation was significantly associated with painful VCF.

## 1. Introduction

After the liver and lungs, bone is the third most common organ where metastasis occurs [1]. Although almost all malignancies can metastasize to the skeleton, 80% of bone metastases originate from the breast, prostate, lung, kidney, and thyroid cancers [2]. Innovations in systemic therapy for many cancer types have prolonged the life expectancy of cancer patients, including those with bone metastases. One of the most common symptoms of bone metastasis is pain, which affects the patient’s quality of life. Based on the results of past large-scale clinical trials, 30–40% of palliative radiotherapy for painful bone metastases was directed to the spine [3,4,5]. Indeed, spinal metastases can cause pain, spinal cord compression, hypercalcemia, and pathologic fracture [6], resulting in decreased quality of life. Conventional external beam radiotherapy has been the gold standard treatment for painful bone metastases with palliative intent [6,7]. A recent phase III trial of the SC.24 showed that the pain-relieving effects of stereotactic body radiotherapy (SBRT) were significantly superior to those of conventional radiotherapy [8]. SBRT for painful spinal metastases, therefore, is established as one of the standard treatments.

Spine SBRT, which can deliver high-dose radiation to the target tissues, leads to excellent tumor control [9]; however, it is associated with a higher risk of vertebral compression fracture (VCF) than conventional radiotherapy [10,11]. A propensity score-matched analysis adjusted for VCF risk factors showed that the SBRT group had a higher 5-year rate of VCF than the conventional radiotherapy group (22% vs. 7%, respectively, *p* = 0.044) [10]. Furthermore, long-term follow-up data from a sub-cohort of patients in the SC.24 randomized trial treated at the Sunnybrook Odette Cancer Center (Toronto, Canada) shows the VCF rate in the SBRT group at 6.7% (8 of 119 spinal segments) and in the conventional radiotherapy group at 2.4% (4 of 169 spinal segments) [11]. 

The VCFs induced by spine SBRT occasionally cause pain and spinal deformity, neurological deficit, spinal instability, or spinal cord compression [12]. Nevertheless, there is a certain number of patients with painless VCF based on the experience of daily clinical practice. As SBRT for painful spinal metastases is performed to relieve pain, the occurrence of symptomatic adverse effects must be avoided. However, most previous studies on VCF caused by SBRT did not report any symptoms of this condition [13,14]; thus, the incidence of painful VCF is unclear. The purpose of the present study was to determine the rate and prognostic factors of painful VCF caused by spine SBRT.

## 2. Materials and Methods

### 2.1. Patients and Data Acquisition

A retrospective review of the medical databases at our institution was conducted to identify the patients treated with SBRT for spinal metastases between August 2013 and December 2021. Patients were included if they met the following criteria: (i) spinal metastases treated with SBRT, (ii) lesions evaluated with magnetic resonance imaging (MRI) or computed tomography (CT) after SBRT, (iii) imaging evaluations revealed a new VCF or progression of an existing VCF in the treated spinal segments, and (iv) pain on the irradiated region was evaluated at the time when VCF occurred after the SBRT treatment.

This study was approved by our institutional ethical review board (approval number: 2035), and informed consent was obtained in the form of an opt-out option displayed on the website.

### 2.2. Spine SBRT

The SBRT technique has been discussed in great detail in previous publications [15,16] and is briefly summarized here. The clinical target volume included the gross tumor and the immediately adjacent bony anatomic compartments at risk of microscopic disease extension, as described by the contouring guidelines for spine SBRT [17,18]. A 2-mm margin was added to the clinical target volume to create the planning target volume (PTV). The prescribed dose (PD) was 20 Gy in a single fraction for curative intent, 24 Gy in two fractions for palliative intent, or 30 Gy in five fractions as a second SBRT course. The planning goal was that 95% of the PTV was irradiated by a dose (D_95%_) that was as close as possible to 100% of the PD under the condition that normal tissues satisfied the dose constraints (PTV D_95%_ ≤ 100% PD). In addition, we set the following constraint for the PTV: the maximum dose should not exceed 140% of the PD (PTV D_max_ ≤ 140% PD) between August 2013 and March 2019 (Figure 1A,B) and 160% of the PD (PTV D_max_ ≤ 160% PD) between April 2019 and January 2020. From February 2020 onward, the maximum dose was set as lower than 170% of the PD (PTV D_max_ ≤ 170% PD).

### 2.3. Definition of VCF

VCF events were defined as the development of a new VCF or the progression of an existing VCF in vertebral bodies, on the basis of imaging evaluations (Figure 1C,D) [19]. All post-SBRT images were reviewed to collect information on the incidence of VCF. First, the pathological VCF that was defined as a fracture with tumor recurrence before or at the time of developing into a VCF was excluded since the purpose of the current study was to assess the pain from VCF caused due to SBRT. The remaining iatrogenic VCFs were classified into three groups: painless (grade 1 according to Common Terminology Criteria for Adverse Events [CTCAE] version 5.0 [20]); mildly painful, needing prescription analgesics (grade 2); and intensely painful, requiring hospitalization or invasive interventions such as percutaneous cement injection or surgery (grade 3). If the patients had VCF at two spinal levels and pain in the area, we counted it as double-painful VCF.

### 2.4. Evaluation and Statistical Analysis

The primary endpoint in the present study was the rate of painful VCF (grades 2–3). In addition, some factors were selected as potential predictors, and their impact on painful VCF was assessed in comparison to that on painless VCF using a univariate model. The factors included age, number of spinal levels (single vs. multiple), spinal instability neoplastic score (SINS) [21], each component of SINS, prior radiation to the treated segment, dose fractions schedules of SBRT [22], the maximum dose of SBRT, de novo fracture vs. progression of an existing fracture, and time interval from SBRT to VCF. Components of SINS included location (junctional, mobile spine, semi-rigid, or rigid), presence or absence of mechanical pain, type of bone lesion (lytic, mixed, or blastic), radiographic spine alignment, vertebral body collapse, and posterolateral tumor involvement [21]. Posterolateral tumor involvement is defined as facet, pedicle, or a costovertebral joint fracture or replacement with a tumor [21]. Extrapolating from the Spine Response Assessment in Neuro-Oncology group recommendation [23], local failure was defined as tumor progression or any new tumors within the epidural space based on MRI or CT and was calculated in months (from the starting date of radiotherapy to the date of tumor progression or the date of the last follow-up for the imaging study). Overall survival was defined as the interval between radiotherapy and the most recent follow-up or death due to any cause. 

Univariate analysis was performed using Fisher’s exact test. Results with *p* < 0.05 were considered statistically significant. Because patient death without tumor recurrence was regarded as a competing risk factor, local failure was estimated using the cumulative incidence function adjusted for the competing risk of death. Overall survival was estimated using the Kaplan–Meier method. All statistical analyses were performed using the EZR software, version 1.54.

## 3. Results

### 3.1. Patient Characteristics

Among the total of 412 patients with 934 spinal segments who underwent SBRT, 391 with 779 spinal segments satisfied the inclusion criteria of this study. The median follow-up for the entire cohort was 18 (range, 1–107) months. The 1-year survival and local failure rates were 83.7% and 8.9%, respectively. 

We observed 19 pathological fractures and 60 iatrogenic fractures (60 of 779, 7.7%) with 31 de novo and 29 that progressed from an existing fracture. Five VCFs occurred at the spinal segment adjacent to the target (these were diagnosed as SBRT-related VCFs based on age, sex, VCF location, no tumor involvement at the vertebra, no osteoporosis, and the time interval from SBRT to VCF). In the cohort of 60 patients with iatrogenic VCFs, the median age was 68 (range, 38–79) years, and 57% of the fractures were observed in women (34 of 60). The most common cancer was thyroid cancer (25%), followed by prostate cancer, sarcoma, colorectal cancer, and lung cancer (12%, 12%, 10%, and 8%, respectively). The number of patients classified as stable (score: 0–6), potentially unstable (score: 7–12), and unstable (score: 13–18) according to SINS was 21, 27, and 12, respectively. Twenty-four (40%) spinal segments had a radiation history and 17 (28%) patients underwent spinal fixation surgery prior to the SBRT. The mean and median time to VCF was 11 and 6 (range, 1–41) months, respectively. Additional patient and tumor characteristics are listed in Table 1.

### 3.2. Painful VCF

Grade 1 (painless), 2 (mildly painful), and 3 (intensely painful or operated) VCFs were confirmed in 41, 4, and 15 segments, respectively. The rate of painful VCF (grade 2–3) was 2.4% (19 of 779) in the spinal segments treated with SBRT. Univariate analysis comparing painful and painless VCF revealed that the painful VCF rate was significantly higher in patients with no posterolateral tumor involvement than in those with bilateral or unilateral involvement (50% vs. 23%; *p* = 0.042) as well as in patients with spine without fixation than in those with fixation (44% vs. 0%; *p* < 0.001). Other investigated factors include age, number of spinal levels, SINS (Stable/Potentially unstable vs. Unstable), SINS components (Location, Pain, Bone lesion, Alignment, and Vertebral body collapse), radiation history, SBRT dose fraction schedules, maximum dose of SBRT, existing VCF, and the interval between SBRT and VCF showed no significant correlation with painful VCF (Table 1).

We have reanalyzed the correlation between painful VCFs and each factor excluding fixation cases for the following two reasons. First, all VCFs were painless in patients who underwent fixation. Second, since SINS is originally used to evaluate lesions prior to fixation, SINS of lesions after fixation might introduce noise. Univariate analysis revealed that none of the investigated factors were significantly associated with painful VCFs (Table 1).

Among 15 spines with grade 3 fractures, salvage surgery including internal fixation for pain from instability or decompression for spinal canal stenosis was performed in 8 patients (8 of 779, 1.0%) (Table 2).

## 4. Discussion

The current study evaluated the pain from VCF caused by the SBRT according to the CTCAE grading system. Sixty-eight percent of iatrogenic VCFs were confirmed to be painless. Although the rate of painful VCFs was only 2.4% of all the treated spinal segments, eight (1.0%) spinal segments required invasive surgery. Univariate analysis showed that the absence of posterolateral tumor involvement and no fixation were significantly associated with painful VCF. 

A large number of studies suggest that SBRT causes VCF more frequently than conventional radiotherapy does [10,11,13,14]. Moreover, most studies have considered the fracture as an event without differentiating between painful and painless VCFs [13,14]. Many factors have been associated with an increased risk of SBRT-related VCF, including the higher dose per fraction, single-fraction SBRT (vs. multi-fraction), pre-existing VCF (vs. new VCF), the presence of a lytic tumor and the associated extent of lytic disease, baseline pain, location in the thoracic spine, a higher pre-treatment SINS, a Bilsky score >0, older age, female sex, and histology (e.g., lung tumor metastases higher, prostate cancer metastases lower) [13,14,24]. 

Symptomatic adverse effects caused by spine SBRT directly affect the patient’s quality of life [25]. Furthermore, salvage interventions for VCF would increase the mental or physical burden on the patient. Therefore, the present study evaluated the rate of symptomatic VCF rather than VCF itself as the primary purpose of this study. Reports on VCF symptoms, including those obtained with different methods of analysis, are summarized in Table 3 [11,19,26,27,28,29,30,31,32,33,34,35]. These reports showed that painful VCF was confirmed in 4.0–21% of the total spinal segments treated with SBRT. The present result of 2.4% was lower than those shown in previous reports [11,19,26,27,28,29,30,31,32,33,34,35]. The discrepancy between previous reports and our results may be attributed to one of the following factors: (i) differences in indications for salvage interventions (only for cases with severe pain from instability or spinal canal stenosis in our institution), (ii) the low dose per fraction (≤19 Gy [22]) in most cases, or (iii) the Japanese patient population with lower body mass.

A phase II trial suggests that prophylactic cement augmentation immediately after SBRT may reduce the risk of VCF (1-year VCF rate of 10%) [36]. However, it is unclear which patient population may benefit the most from this invasive procedure. The secondary purpose of the current study was to clarify which population was at high risk of painful VCF. The absence of posterolateral tumor involvement and absence of fixation before SBRT were found to be significantly correlated with painful VCF in the univariate analysis. Some studies reported irreproducible results regarding various risk factors associated with painful VCF, including high SINS [29], lack of bisphosphonate usage [33], greater PTV D90%, de novo kyphosis or scoliosis, and vertebral body collapse [35] (Table 3). In general, VCF risk factors can be divided into three types: patient characteristics (age, sex, hormone imbalance, physique, etc.), lesion factors (SINS and its components), and external factors (dose fraction schedule, radiation dose per fraction, irradiated field, and dosimetric parameters). Due to the large number of confounding factors involved, it may be difficult to obtain reproducible results. 

To clarify the predictors of painful VCF in the present study, a univariate analysis was performed to compare between patient population with painful VCF and those with painless VCF. An alternative study design was also considered for data analysis, which involved comparing patients with painful VCF to all the remaining patients treated with SBRT. However, we did not adopt the latter study design because, in such analysis, it was anticipated that the prognostic factors of the VCF, as reported in previous studies [13,14,24], would exhibit significant differences. As a result, this study identified risk factors for painful VCF that differed from those associated with VCF in general, either painless or painful.

The reason why VCF induced by SBRT is often painless remains unclear. However, the current findings, which demonstrate painless VCFs in patients with fixation prior to SBRT, suggest that the cause of pain in painful VCFs is associated with spinal instability. The present study also revealed that the absence of posterolateral tumor involvement was significantly associated with painful VCFs. This might be due to the severe deterioration of spinal alignment when VCF occurred in the posterolateral robust spine. However, in a repeat analysis that excludes fixation cases, the absence of posterolateral tumor involvement was not associated with painful VCFs.

This study has several strengths. First, this is one of the few studies that is focused on symptoms caused by VCF. We argue that it is not the incidence of VCFs which is of primary importance, but rather the symptoms caused by the VCF. Second, this study only included patients who were treated with spine SBRT of three uniform dose fraction schedules. In addition, 24 Gy in two fractions which is the standard dose for pain palliation [8] was selected in 78.5% (307/391) of cases. Despite the strengths, there are also limitations to this study. First, the analyzed cohort was heterogeneous, and there were differences in potential predictors such as original bone density and history of systemic therapy. Second, the study evaluated a small number of painful VCF events; therefore, the results of the univariate analysis might have false positive and negative results. To confirm the reproducibility of the results, large-scale clinical studies are required in the future.

## 5. Conclusions

The present study showed that painful VCF occurred in only 2.4% of the spinal segments treated with SBRT. In contrast, salvage surgery was needed in 1.0% of spinal segments, which is useful information when we explain spine SBRT to the patients. Additionally, the absence of posterolateral tumor involvement and no fixation were suggested to be the risk factors of painful VCF.

## Figures and Tables

**Figure 1 jcm-12-03853-f001:**
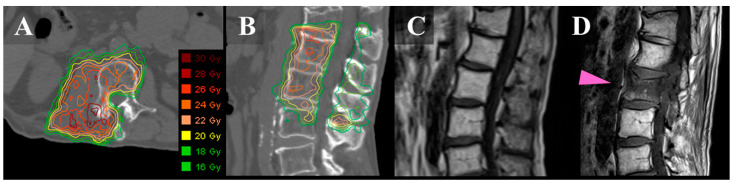
Images of a 67-year-old man with metastatic L-1/2 sarcoma. Axial (**A**) and sagittal (**B**): CT images with dose distribution of SBRT. The spinal lesions show a right posterior involvement. (**C**): T1-weighted sagittal MRI before SBRT. (**D**): T1-weighted sagittal MRI 3 months after SBRT showing de novo VCF in the second lumber spine (pink arrow). The VCF did not cause pain. CT, computed tomography; MRI, magnetic resonance imaging; SBRT, stereotactic body radiotherapy; and VCF, vertebral compression fracture.

**Table 1 jcm-12-03853-t001:** Patient and tumor characteristics and univariate analysis to identify factors associated with painful fractures.

Characteristic		All Lesions			Lesions Excluding Fixation Cases		
		No. of Lesions (*n* = 60)	Painful VCF	*p* Value	No. of Lesions (*n* = 43)	Painful VCF	*p* Value
Age, years	≤65 (range, 38–65)	28	8	0.782	19	8	1.000
	>65 (range, 67–79)	32	11		24	11	
Sex	Male	26	10	0.405	17	10	0.209
	Female	34	9		26	9	
Lesion histopathology	Thyroid	15	4	NA	9	4	NA
	Prostate	7	5		7	5	
	Sarcoma	7	1		5	1	
	Colorectal	6	3		5	3	
	Lung	5	0		4	0	
	Breast	4	1		3	1	
	Renal cell	4	1		2	1	
	Others (1–2 cases each)	12	4		8	4	
Number of spinal levels	1 2/3/4/5+	33 15/4/5/3	12 7	0.419	23 20	12 7	0.359
SINS	Stable/Potentially unstable Unstable	48 12	16 3	0.735	34 9	16 3	0.708
Location	Occiput–C2, C7–T2, T11–L1, L5–S1 C3–6, L2–4 T3–10 S2–5	27 20 13 0	10 8 1 0	0.096	21 16 6 0	10 8 1 0	0.411
Pain	+ Occasional pain but not mechanical -	23 11 26	5 3 11	0.313	17 6 20	5 3 11	0.285
Bone lesion	Lytic Mixed Blastic	37 11 12	11 4 4	0.926	24 7 12	11 4 4	0.591
Spinal alignment	Subluxation/translation present De novo deformity Normal	11 9 40	4 2 13	0.843	9 3 31	4 2 13	0.875
Vertebral body collapse	>50% collapse <50% collapse No collapse with >50% body involved None	12 19 8 21	1 8 1 9	0.087	6 16 4 17	1 8 1 9	0.413
Posterolateral tumor involvement	Bilateral/Unilateral None	40 20	9 10	0.042	24 19	9 10	0.368
Radiation history to the treated spine	+ -	24 36	7 12	0.784	17 26	7 12	1.000
Fixation before SBRT	+ -	17 43	0 19	<0.001	NA	NA	NA
Prescribed dose of SBRT	20 Gy/1 fx 24 Gy/2 fx 30 Gy/5 fx (as a 2nd SBRT course)	4 49 7	1 15 3	0.863	4 34 5	1 15 3	0.646
Maximum dose of SBRT	≤140% prescribed dose 140% <, ≤160% prescribed dose 160% <, ≤170% prescribed dose	27 16 17	8 5 6	0.938	18 11 14	8 5 6	1.000
VCF	De novo progression	27 33	9 10	0.872	20 23	9 10	1.000
Time interval between SBRT and VCF	1–6 months ≥7 months	31 29	10 9	1.000	26 17	10 9	0.531

fx, fraction(s); NA, not available; SBRT, stereotactic body radiotherapy; SINS, spinal instability neoplastic score; and VCF, vertebral compression fracture.

**Table 2 jcm-12-03853-t002:** Patients who underwent invasive surgery.

Age (Years)/Sex	Type of Cancer	SINS	Prescribed Dose of SBRT (Maximum Dose)	VCF Site	Time to Onset (Months)	Reasons for Surgery	Surgical Form
65/Female	Sigmoid Colon	16	24 (31.1) Gy in 2 fx to C6–Th1 level	C7	0 (8 days)	Spinal canal stenosis	Decompression and fixation
65/Male	Prostate	6	24 (27.6) Gy in 2 fx to L4 level	L4	22	Vertebral canal stenosis	Decompression and fixation
63/Male	Prostate	2	24 (27.8) Gy in 2 fx to L3 level	L3	34	Spinal canal stenosis	Decompression and fixation
58/Male	Bladder	8	24 (38.4) Gy in 2 fx to Th10 level	Th10	5	Severe pain from instability	Fixation
58/Male	Prostate	4	24 (39.7) Gy in 2 fx to S1 level	S1	22	Severe pain from instability and sacral nerve compression	Fixation
58/Male	Prostate	3	24 (28.6) Gy in 2 fx to L3 level/30 (49.8) Gy in 5 fx to L3 level	L3	63/22	Severe pain from instability	Fixation
72/Female	Thyroid	5	24 (40.6) Gy in 2 fx to L1 level	L1	3	Spinal canal stenosis	Decompression and fixation
74/Male	Esophagus	2	24 (40.1) Gy in 2 fx to L4–5 level/30 (51) Gy in 5 fx to L4–5 level	L3	20/4	Spinal canal stenosis	Decompression

fx, fractions; SINS, spinal instability neoplastic score; SBRT, stereotactic body radiotherapy; and VCF, vertebral compression fracture.

**Table 3 jcm-12-03853-t003:** Literature review according to VCF criteria.

Author (Year)	No. of pts/Segments	Median SBRT Dose (Range)	No. of VCF	VCF				Risk Factors of Painful VCF (Statistical Method)
				Gr. 1	Gr. 2	Gr. 3	Pathological	
Cunha et al. (2012) [19]	90/167	20–27 Gy/3 fx (8–35 Gy/1–5 fx)	19 (11.4%)	10 (6.0%)	NA	9 (5.4%)	NA	NA
Sung et al. (2014) [26]	72/72	(18–45 Gy/1–5 fx)	26 (36%)	11 (15%)	NA	15 (21%)	NA	NA
Thibault et al. (2014) [27]	37/61	24 Gy/2 fx (18–30 Gy/1–5 fx)	10 (16%)	6 (10%)	NA	4 (7%)	NA	NA
Germano et al. (2016) [28]	79/143	18 Gy/1 fx (10–18 Gy/1 fx)	30 (21.0%)	NA	NA	8 (6.5%)	8 (6.5%) or more	NA
Lee et al. (2016) [29]	79/100	18 Gy/1 fx, 27 Gy/3 fx (16–27 Gy/1–3 fx)	32 (32%)	17 (17%)	NA	15 (15%)	NA	High SINS (MA)
Virk et al. (2017) [30]	323/551	24 Gy/1 fx	NA	NA	NA	56 (10.2%)	21 (3.8%) or more	NA
Ling et al. (2018) [31]	43/70	16 Gy/1 fx (12–24 Gy/1 fx)	9 (13%)	1 (1%)	2 (3%)	6 (9%)	NA	NA
Mantel et al. (2019) [32]	56/61	35 Gy/5 fx, 48.5 Gy/10 fx	21 (34%)	16 (26%)	2 (3%)	3 (5%)	NA	NA
Ozdemir et al. (2019) [33]	78/125	16 or 18 Gy/1 fx	5 (4.0%)	0	0	5 (4.0%)	NA	Lack of bisphosphonate usage (UA)
Chen et al. (2020) [34]	193/302	24 Gy/3 fx (15–30 Gy/1–5 fx)	38 (12.6%)	Part of 11	Part of 11	15 (5.0%)	12 (4.0%)	NA
Zeng et al. (2022) [11]	66/119	24 Gy/2 fx	NA	3 (2.5%)	0	5 (4.2%)	NA	NA
Zeng et al. (2023) [35]	159/301 323/646	28 Gy/2 fx 24 Gy/2 fx	37 (12.3%) 75 (11.6%)	NA NA	NA NA	15 (5.0%) 35 (5.4%)	NA NA	Greater PTV D90%, de novo kyphosis or scoliosis, vertebral body collapse (MA)
Present study	391/779	24 Gy/2 fx (20 Gy/1 fx, 30 Gy/5 fx)	79 (10.1%)	41 (5.3%)	4 (0.5%)	15 (1.9%)	19 (2.4%)	No posterolateral tumor involvement, no fixation (UA)

D90%, dose irradiated to the 90%; fx, fraction(s); MA, multivariate analysis; NA, not applicable; PTV, planning target volume; SBRT, stereotactic body radiotherapy; SINS, spinal instability neoplastic score; UA, univariate analysis; and VCF, vertebral compression fracture.

## Data Availability

The datasets used and/or analyzed during this study are available from the corresponding author upon reasonable request.

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
