# Peer review of "Incidence and Prognostic Factors of Painful Vertebral Compression Fracture Caused by Spine Stereotactic Body Radiotherapy"

_jcm, 2023, doi:10.3390/jcm12113853_

Round 1
Reviewer 1 Report
Summary
The authors present a retrospective series evaluating the incidence of painful VCF development in 412 patients undergoing spinal SBRT for spinal metastatic disease.
Strengths
The study is relevant to the field. The manuscript is overall well-constructed
Weaknesses
1. Grammatical/spelling errors are present
a. In the Results section, the phrase, “…that progressed form an existing fracture…” should read, “…that progressed from an existing fracture…”.
2. It is unclear to me what is meant by “posterolateral tumor involvement”. A simple figure illustrating the locations of tumor used for these different classifications would be useful. Also, it is unclear why the absence of tumor involvement in the posterior aspect of the vertebral body would make it less likely to have pain associated with a VCF. The authors submit that, “This might be due to the severe deterioration of spinal alignment when VCF occurred in the posterolateral robust spine.”, but wouldn’t “severe deterioration of spinal alignment” be more likely to lead to pain? Overall I think this element needs to be clarified throughout the paper.
3. You note that 5 VCFs occurred adjacent to the SBRT target. Can these VCFs then truly be said to be a result of the SBRT?
4. Was tumor histology evaluated as a potential factor related to the development of painful VCFs? In other words, were certain tumor types more likely to lead to painful VCF than others?
5. In the introduction and/or Discussion, I would recommend that the prior relevant literature be elaborated on. You mention that prior literature has demonstrated an increased risk of VCF in patients undergoing spinal SBRT. What is the odds ratio or increased likelihood of VCF in these studies?
Minor revision to spelling/grammar
Author Response
Please see the attachment.
RESPONSES TO REVIEWER’S COMMENTS
We would like to sincerely thank the reviewer for taking time to review our manuscript and provide insightful comments and suggestions. We have revised the manuscript based on the comments, which have helped to improve the manuscript considerably. Below are our point-by-point responses to the reviewer’s comments and suggestions.
**************************************************************************
Reviewer 1
Summary
The authors present a retrospective series evaluating the incidence of painful VCF development in 412 patients undergoing spinal SBRT for spinal metastatic disease.
Strengths
The study is relevant to the field. The manuscript is overall well-constructed
Weaknesses
- Grammatical/spelling errors are present
- In the Results section, the phrase, “…that progressed form an existing fracture…” should read, “…that progressed from an existing fracture…”.
Response: We have revised it according to the reviewer’s suggestions (line 145, page 4). In addition, the whole manuscript was rechecked by a native English speaker and a proofreader (we have attached the certification).
- It is unclear to me what is meant by “posterolateral tumor involvement”. A simple figure illustrating the locations of tumor used for these different classifications would be useful.
Response: Thank you for this constructive suggestion. The “posterolateral tumor involvement” is one of the SINS components [21]. However, as pointed out by the reviewer, it was difficult to understand this term. Therefore, in order to clarify its meaning, we have included the definition of posterolateral tumor involvement as "facet, pedicle, or a costovertebral joint fracture or replacement with tumor [21]" in the Materials and Methods section (line 124-126, page 3).
Also, it is unclear why the absence of tumor involvement in the posterior aspect of the vertebral body would make it less likely to have pain associated with a VCF. The authors submit that, “This might be due to the severe deterioration of spinal alignment when VCF occurred in the posterolateral robust spine.”, but wouldn’t “severe deterioration of spinal alignment” be more likely to lead to pain? Overall, I think this element needs to be clarified throughout the paper.
Response: We apologize to have caused your misunderstanding. The findings showed that the absence of posterolateral tumor involvement was significantly associated with painful VCF.
- You note that 5 VCFs occurred adjacent to the SBRT target. Can these VCFs then truly be said to be a result of the SBRT?
Response: Thank you for the important remarks. We consider that multiple factors were contributing in combination, but based on the following evidence, we determined it to be SBRT-related VCF.
A pathological fracture was considered to be ruled out because there is no tumor in the vertebral body. Based on age, sex, VCF location, and no osteoporosis, fragility fractures were considered to be negative.
SBRT-related fracture was most suspected with reference to the interval between SBRT and VCF.
We added it in the Results section as follows: “these were diagnosed as SBRT-related VCFs based on age, sex, VCF location, no tumor involvement at the vertebra, no osteoporosis, and the interval from SBRT to VCF.” (line 146-148, page 4)
- Was tumor histology evaluated as a potential factor related to the development of painful VCFs? In other words, were certain tumor types more likely to lead to painful VCF than others?
Response: Due to many types of cancer and small sample sizes for each, it was not possible to perform a significant difference test.
- In the introduction and/or Discussion, I would recommend that the prior relevant literature be elaborated on. You mention that prior literature has demonstrated an increased risk of VCF in patients undergoing spinal SBRT. What is the odds ratio or increased likelihood of VCF in these studies?
Response: We agree that the information the reviewer pointed out should be written in the Introduction of the paper.
We added the following sentences (line 50-56, page 2): A propensity score-matched analysis adjusted for VCF risk factors showed that the SBRT group had a higher 5-year rate of VCF than the conventional radiotherapy group (22% vs. 7%, respectively, p = 0.044) [10]. Furthermore, long-term data from a sub-cohort of patients in the SC.24 randomized trial treated at the Sunnybrook Odette Cancer Center (Toronto, Canada) shows the VCF rate in the SBRT group at 6.7% (8 of 119 spinal segments) and in the conventional radiotherapy group at 2.4% (4 of 169 spinal segments) [11].
The odds ratio in the latter study was 2.97.

Reviewer 2 Report
Authors performed a review of painful vertebral compression fractures after SBRT. The study is interesting but several items need to be improved/clarified
1. Authors should consult an english-editing service as the syntax is very confusing and manuscript is difficult to read
2. The timepoint at which pain was assessed needs to be better clarified. Was this at any time during the follow-up period? Was this at 2 weeks, 6 weeks postop?
3. Is there data on pain scores before SBRT?
Language needs to be improved. Would recommend an English editing service.
Author Response
Please see the attachment.
RESPONSES TO REVIEWER’S COMMENTS
We would like to sincerely thank the reviewer for taking time to review our manuscript and provide insightful comments and suggestions. We have revised the manuscript based on the comments, which have helped to improve the manuscript considerably. Below are or point-by-point responses to the reviewer’s comments and suggestions.
**************************************************************************
Reviewer 2
Comments and Suggestions for Authors
Authors performed a review of painful vertebral compression fractures after SBRT. The study is interesting but several items need to be improved/clarified
- Authors should consult an english-editing service as the syntax is very confusing and manuscript is difficult to read
Response: We apologize for the inconvenience. The whole manuscript was revised by a native English speaker and a proofreader (we have attached the certification).
- The timepoint at which pain was assessed needs to be better clarified. Was this at any time during the follow-up period? Was this at 2 weeks, 6 weeks postop?
Response: Thank you for your suggestion. We agree that the timing of pain evaluation should be clarified. VCF was graded based on the exacerbation of pain observed when VCF occurred.
We added the information on Material and Methods section as follow: Pain on the irradiated region was evaluated at the time when VCF occurred after the SBRT treatment (line 73, page 2).
- Is there data on pain scores before SBRT?
Response: The information on the pain scores before SBRT was not available from the medical records; however, we assumed that it was within the tolerance limits, otherwise, the patients should have complained.

Round 2
Reviewer 1 Report
Meets criteria for publication in my opinion.
Reviewer 2 Report
Authors have revised satisfactorily.